## Case Study

adaptation; suicide; trauma; refugee; psychotherapy

**Corresponding author:**
Srishti Meera Sardana;
Email: sardana@uwm.edu

# A suicide safety protocol with vulnerable populations with chronic physical health conditions: A pragmatic protocol implementation among displaced Myanmar adults in Thailand

Srishti Meera Sardana[1,2] , Ye Htut Oo[3,8], Khin Nyein Chan Soe[5], Htin Aung[5], Stephanie Van Wyk Skavenski[1], Amanda Nguyen[6] , Laura K. Murray[1], Jarntrah Sappayabanphot[4], Judith K. Bass[1] , Catherine Lee[3,7], Wongsa Laohasiriwong[8] and Kaung Nyein Aye[4]

[1]Johns Hopkins Bloomberg School of Public Health, USA; [2]Psychological and Brain Sciences, University of Wisconsin-Milwaukee, USA; [3]International Rescue Committee, USA; [4]International Rescue Committee, Thailand; [5]CETA Global, Thailand; [6]School of Education and Human Development, University of Virginia, USA; [7]Community Partners International, USA and [8]Khon Kaen University, Thailand

## Abstract

Suicide is a serious global health problem with ~73% deaths by suicide occurring in low- or middle-income countries (LMICs), many of which are among people experiencing humanitarian emergencies. Few guidelines outline specific steps and strategies to tackle suicide risk and manage post-attempt consequences in these settings, leaving program implementers with limited information to translate guidance to practice. This article describes the implementation of the Common Elements Treatment Approach (CETA) suicide safety protocol as part of a randomized controlled trial in northern Thailand for displaced adults from Myanmar with chronic physical health conditions. The CETA safety protocol has been used in many trials and programs to screen for and manage suicide risk, including in a prior CETA effectiveness trial with Myanmar adults. In this article, we describe how this safety protocol was adapted for the study setting, and utilized to effectively screen, assess suicidal thoughts and behaviors, develop and manage action plans for study participants with active suicidal ideation. We present three illustrative case descriptions of individuals with whom we implemented the safety protocol to highlight how suicide risk intersects with physical illness, psychosocial stressors and structural vulnerability. Reflections on feasibility, acceptability and adaptations – such as language translation, culturally grounded referral pathways and training for nonspecialist providers – are shared to inform future implementation. Our findings support the implementation of suicide safety protocols within humanitarian programming and offer practical insights for global health practitioners and policymakers working in similarly complex settings.

## Impact statement

Suicide prevention remains an overlooked yet urgent need in humanitarian settings, particularly among individuals managing chronic physical health conditions alongside social and emotional distress. This study offers a grounded, field-based perspective on how suicide risk can be effectively identified and addressed using the Common Elements Treatment Approach safety protocol, a flexible and scalable tool that can be implemented as part of routine service delivery in low-resource contexts. Drawing on a real-world example with displaced Myanmar adults in northern Thailand, this study illustrates how trained providers can screen for suicide risk, develop tailored safety plans and coordinate care – even in settings with limited mental health infrastructure. The inclusion of case examples brings visibility to the lived realities of people navigating displacement, poverty and illness, underscoring how these intersecting challenges heighten vulnerability to suicide. Rather than presenting a prescriptive model, this work shares practical insights into what implementation looks like on the ground – including common barriers, ethical dilemmas and adaptations made in response to cultural, contextual and procedural considerations. These reflections offer valuable guidance for those designing or delivering mental health and psychosocial support services in similarly complex

environments. Ultimately, this study encourages a shift in thinking: suicide prevention is not a specialized or standalone intervention, but a critical component of holistic, person-centered care. By embedding safety protocols within broader health programming, practitioners and policymakers can make meaningful strides toward addressing mental health needs in ways that are responsive, feasible and sustainable.

## Background

Suicide is a global public health problem. More than 7,20,000 people die by suicide each year (WHO, Suicide Worldwide, 2025), that approximates to one suicide-related death every 40 s. Approximately 73% of deaths by suicide occur in low- or middle-income countries (LMICs), many of which are among people experiencing humanitarian emergencies (IASC Suicide Guidance Note, December, 2022). Reducing suicide mortality is a targeted United Nations Sustainable Development Goal (UN SDG, 2023).

There are a few general guidance documents that address risk and post-attempt consequences of suicide (Lund et al., 2018). The Inter-Agency Standing Committee: Mental Health and Psychosocial Support Reference Group suicide guidance recognizes three dimensions (individual, relationships and community/society) to consider while developing and implementing suicide prevention strategies in humanitarian emergencies (IASC MHPSS Suicide Guidance Note, December, 2022). The UNHCR guidance suggests six steps to develop an action plan for suicide prevention and mitigation (UNHCR, 2023), which include creating a multisectoral taskforce (Step 1); collecting, analyzing and synthesizing information (Steps 2 and 3); prioritizing and selecting interventions (Step 4); ensuring key personnel buy-in (Step 5); and developing a system to track outcomes (Step 6) (UNHCR, 2023). Both documents advise on general approaches and suggest that strategies should be tailored to the needs of high-risk, vulnerable groups, but neither document provides specific steps and implementation instructions.

The interplay between chronic physical illnesses and mental health conditions is often underrecognized and inadequately addressed (Mateen, 2022). Common chronic illnesses, such as diabetes and hypertension, not only pose significant physiological challenges but can also inflict considerable psychological burdens (Berk et al., 2023). In humanitarian emergencies, people with chronic conditions, such as diabetes and cardiovascular disease, experience increased vulnerability due to disrupted healthcare services, medication shortages and heightened psychosocial stressors (Schmid and Raju, 2021; Asgary et al., 2022; Ayeb-Karlsson et al., 2023). Social isolation, loss of support networks and compromised access to essential medications can exacerbate psychological distress, rendering them acutely susceptible to feelings of hopelessness and despair (Ayeb-Karlsson et al., 2023). With studies estimating that chronic disease prevalence among displaced populations can be as high as 20–30%, consideration of mental health needs of these populations is particularly important (IOM, 2018; Fine et al., 2022). Addressing risk for suicide and preventing suicide among this population requires context-appropriate protocols that can improve suicide assessment and management.

This article describes the implementation of a previously described suicide safety protocol (Murray et al., 2014) from the Common Elements Treatment Approach (CETA) that has been used in numerous trials and a range of humanitarian settings (see, e.g., Bonilla-Escobar et al., 2018; Murray et al., 2020; Bogdanov et al., 2021). CETA is a flexible, evidence-based, modular transdiagnostic mental health intervention approach that is designed to address multiple co-occurring mental health conditions (Murray et al., 2013). It is typically provided on an individual basis through weekly 1-h sessions. CETA can be delivered via phone or in person at a clinic or during home visits. The CETA intervention draws on a set of common cognitive-behavioral and interpersonal strategies, such as psychoeducation, cognitive restructuring, behavioral activation, relaxation training, problem-solving and exposure techniques. Counselors are trained to select and sequence these elements based on client presentation and evolving needs, which allows for tailoring while maintaining fidelity to the model. Supervision and structured decision-making tools are embedded in the approach to support quality delivery, particularly when provided by nonspecialist providers in humanitarian or low-resource contexts. The CETA program implemented in this study was adapted and evaluated as part of a prior efficacy trial among displaced persons on the Thailand/Myanmar border (Bolton et al., 2014). In this article, we describe the protocol in detail, adaptations made for the displaced persons context and provide three individual case examples with whom we implemented the safety protocol to assess and manage their suicidality risk. We reflect on lessons learned in the setup and implementation of the protocol, as key recommendations from this approach may be useful for researchers and program staff who want to implement suicide screening and management programming.

## Program context

Over the past decade, the number of refugees and displaced people in the Asia-Pacific region has increased by 3%. As of 2023, Thailand is hosting more than 90,000 displaced Myanmar nationals in temporary shelter communities on the Thai-Myanmar border (UNHCR Fact Sheet, 2023). The International Rescue Committee (IRC) has a comprehensive primary healthcare program for displaced Myanmar people living in Tha Song Yang, a district on the Thai-Myanmar border. Displaced persons living in these settlements face numerous challenges, including overcrowded living conditions, limited privacy, restrictions on movement and employment, prolonged displacement with no durable solutions and limited access to education and healthcare services. These factors contribute to ongoing psychosocial stress and increase vulnerability to mental health disorders and suicide risk (Handicap International, 2016).

With a grant from the Enhancing Learning and Research for Humanitarian Assistance under the Research for Health in Humanitarian Crises funding call, the IRC, together with faculty from Khon Kaen University (KKU) in Thailand and the Johns Hopkins University in the United States, implemented a randomized controlled trial (RCT) using CETA (Murray et al., 2013) to address poor medication adherence among people with diabetes and/or hypertension, with and without comorbid mental health problems. Ethical approval for the RCT was obtained from KKU (IRB 00008614). This approval included all study procedures, primary and secondary outcomes assessments and implementation of the safety protocol. The RCT in which the suicide safety protocol was implemented was launched in September 2022 in the Mae Sot displaced persons settlement. Poor (<70%) medication adherence to their diabetes and/or hypertension medication was the primary outcome for the RCT, along with assessing HbA1c and fasting

plasma glucose (for participants with diabetes) and blood pressure (for participants with hypertension). All study participants were assessed for mental health symptoms and substance use behaviors. Assessments were completed at baseline and at 3- and 6-month follow-ups (trial manuscript under review).

## Suicide safety protocol

The suicide safety protocol was intentionally designed to be feasible for nonmental health specialist providers, incorporating low-burden tools and a clear decision tree to support real-time supervision and decision-making. The protocol draws upon the Interagency Steering Committee (IASC MHPSS Suicide Guidance Note, December, 2022) guidelines and United Nations High Commissioner for Refugees (UNHCR) steps (UNHCR, 2023). Specifically, the IASC guides practitioners to adopt approaches that leverage the presence of family/friends/relatives. In our safety planning for high-risk cases requiring the involvement of family members to maintain safety watch (as described in Cases 1 and 2 below), we utilized this same approach effectively. In line with the UNHCR's six steps of suicide management, we worked with the CETA counselors to systematically collect, assess and synthesize screening and assessment data to inform safety planning (Steps 2–3) and to cultivate coping strategies to manage distress and address suicidal thoughts or behaviors (Step 4). The protocol also establishes regular check-ins and alignment with the CETA counselors, supervisors and trainers, as well as the community-based psychosocial unit crisis management staff (Step 5), and developed a robust suicide risk management and outcomes system (Step 6).

The CETA suicide safety protocol includes three main components: (1) suicide risk screening, (2) level of risk assessment and (3) suicide risk management. The suicide risk screening was administered to all study participants at all data collection time points and for participants in the CETA arm, at each CETA session. During data collection, study staff completed the screening and tracked participant response; during the CETA program, CETA counselors completed the screening. The screening component of the protocol begins with three questions assessing initial risk for suicide, homicide and domestic violence (Table 1). While this article focuses on suicide risk, the broader safety protocol addressed all three domains (Murray et al., 2014).

If a participant scored >1 on any of the screening questions, thereby disclosing suicidal thoughts and/or behaviors during data collection, the study staff contacted a CETA counselor to implement follow-up actions, including completing a safety plan. If the

**Table 1.** Initial safety risk assessment

|  | Score scale | | | |
| --- | --- | --- | --- | --- |
|  | None of the time | A little of the time | Most of the time | All of the time |
| How often have you had thoughts of wanting to hurt or kill yourself | 0 | 1 | 2 | 3 |
| How often have you had thoughts of wanting to hurt or kill others? | 0 | 1 | 2 | 3 |
| How many times did your partner threaten to hurt you? | 0 | 1 | 2 | 3 |

*Note*: The time frame varied: At baseline – in the last 2 weeks; at treatment session – since the last session.

participant scored >1 during a CETA session, the counselor completed the follow-up actions directly. Participants who scored >1 on the suicide screening question were asked four follow-up questions to assess risk level. Based on standard protocols (Mann et al., 2005; Stanley and Brown, 2012), safety risk was ascertained at *low* risk (thoughts only), *medium* risk (thoughts and plan/method) and *high* risk (thoughts, plan/method and means). If the participant was identified as *low* risk, the CETA counselor contacted their supervisor at the end of the session. If the participant was at *medium* or *high* risk, the counselor immediately contacted the supervisor in the presence of the participant; if the CETA counselor was unsure of the risk level, they were trained to call the supervisor into the session. In all cases, the CETA counselor presented the participant's responses to the supervisor during regular weekly supervision sessions and discussed the proposed safety plan, steps for oversight and follow-up plans. The case information and safety plans were recorded in a clinical tracking form for program monitoring. CETA counselors recorded screening results and safety risk level as part of regular CETA monitoring within 24 h of completing the CETA session, which was then reviewed by the CETA supervisor. Since this was implemented as part of an RCT, the session notes were also discussed with CETA trainers during regular study oversight calls.

A typical safety plan involved sequential steps for risk mitigation and could include: identifying warning signs and providing skills to manage safety risk; deploying a safety contract (i.e., participant makes a promise [verbally or in writing] to keep herself safe for the next 24 h); and/or activating a safety watch (i.e., when a trusted individual keeps a watch to ensure safety for 24 h). A safety plan was developed for all participants who indicated any elevated risk (i.e., scored >1 on the initial screening questions in Table 1). The specific safety plan for each at-risk participant was designed collaboratively with the participant to ensure it was practical, personalized and feasible for them. The participant's own resources, such as their cell phone and family or community contacts, were incorporated to develop supports. The persons that the participant perceived as most appropriate and accessible (e.g., spouse or a religious leader at the local Mosque) were included in the safety plan. The development of the safety plan typically took 20–30 min of one-on-one time to engage the participant and support their comfort in discussing their suicide risk and safety plan. CETA counselors used therapeutic techniques that were practiced with their supervisor, including reflective listening, asking open-ended questions, to help the participant problem-solve and identify barriers.

## Adaptation of the suicide safety protocol

We adapted the CETA safety protocol delivery to be in line with existing resources and referral systems, and to be consistent with the local norms. The protocol was already available in Burmese and Karen languages from the previous trial (Bolton et al., 2014). To ensure continued appropriateness, the safety protocol was reviewed by local IRC staff and their psychosocial crisis management teams, and in consultation with the newly trained CETA supervisors. To make the safety plan implementable, the types of locally available resources and sources of support were identified in these discussions.

Given limited specialized mental health resources, for participants reporting elevated suicide risk and needing a safety plan, the follow-up was conducted by the CETA counselors and supervisors directly. This included safety checks implemented through each study follow-up and connecting participants with the IRC-supported primary care clinics and community health worker

networks. When participants could not be reached by phone or missed their clinic visits, home visits were conducted. For participants with limited family and social supports, CETA counselors engaged community health workers as part of safety planning. Daily safety checks were implemented for those with elevated risk, ensuring continuous, context-appropriate support.

Based on learnings recorded in the CETA clinical monitoring and supervision system, one further adaptation to the protocol included implementing the safety check-ins for those with moderate/high risk in conjunction with medication adherence checks. This was particularly relevant in this study because participants with hypertension and/or diabetes with comorbid anxiety, depression or post-traumatic stress disorder symptoms reported forgetting to take their medication and difficulty remembering to follow safety steps.

## Results

A total of 224 adults completed the RCT baseline assessment, which included the screening component of the safety protocol. Thirteen (5.80%) scored a 1 or higher on the *thoughts of wanting to hurt or kill yourself* question, requiring follow-up suicide safety assessment. Among the 13 participants (1 male and 12 females), 5 had hypertension, 3 had diabetes and 5 had both conditions. At the baseline assessment, nine participants screened at low, two at moderate and two at high risk for suicide. Below, we provide case examples of three participants with moderate/high suicide risk. These cases were selected in discussion with the local team to demonstrate the interaction of suicide risk with the two chronic illness conditions included in the study: diabetes and hypertension.

### Case 1: Karen – A 66-year-old female with hypertension

At the initial safety risk assessment, the participant reported having suicidal thoughts occurring three times a week and was overwhelmed and distressed. She described the initiation of her distress as being related to her money and belongings being recently stolen at the market where she was selling things. This participant presented with co-occurring depression symptoms and reported smoking ~35 cigarettes a week. She described her plan to buy sleeping pills or use a knife to kill herself and identified a place where she could purchase the pills from (thereby articulating thoughts, plan and having the means to kill herself), confirming her 'high risk' status. She reported no prior history of suicide. A recent widow, she was living with her daughter and eight other family members, many of whom depended on her for financial support.

She reported increased nighttime suicidality with unhelpful thoughts including '*It would be better if I died*', '*I am the only one struggling for the whole family*', '*I am a widow and cannot rely on anyone*' and '*my children are useless and could not even arrange a funeral for me if I died*'. The counselor developed a safety plan that included a safety contract, a safety watch for her to be monitored by her daughter and twice daily phone check-ins to ensure she was safe. The safety plan also involved spending time with friends and grandchildren, and going to church for her prayer meetings, where she reported finding peace and solace. This participant's CETA treatment focused on helping her develop cognitive coping skills, learning to restructure problematic thoughts, using behavioral activation strategies to address depressive symptoms and using progressive muscle relaxation techniques to offset nighttime

worries about her problems. The counselor also worked with her to develop problem-solving skills to help the participant find contingency employment opportunities, as well as come up with a realistic plan to pay off her debts.

This participant's safety risk dropped to a 'low' level after 1 week of implementing the safety plan. Due to the initial high risk, the counselor continued to follow up every 3 days to ensure her safety for the first month of CETA treatment, to monitor her status in between regularly scheduled CETA sessions. During the weekly CETA sessions, the counselor continued to assess safety. Over 18 CETA sessions, the participant learned to restructure her unhelpful thoughts to helpful ones about her future, health, ability to work and her faith. She also showed high motivation to manage her hypertension with regular medication intake and learned to cope with daily situations by using the skills she had learned. There was no relapse to suicidality as assessed at the final study follow-up (6-month follow-up), and her improvement was reflected in her end-of-treatment drop in depression symptom severity and reduction in cigarette usage (down to seven cigarettes a week).

### Case 2: Karen – A 46-year-old female client with diabetes and hypertension

At the initial presentation, the participant reported symptoms of post-traumatic stress and depression with 'high risk' for suicide. She had hypertension and diabetes and a history of traumatic loss when she had lost her newborn baby 17 years ago. The participant reported still experiencing recurring thoughts about the baby's death. She had been dealing with thoughts of suicide for the past 4 years, including specific plans to hang herself using a rope or drink acid, as she had attempted 4 years ago. In establishing her safety plan, the participant's husband was brought in to help remove potential dangers. He involved her other relatives, with whom the participant was comfortable disclosing her suicidality, to create a safety watch. As part of the safety protocol, due to the intensity of her symptoms and imminent danger to self and others, the CETA counselor also liaised the participant with a psychiatrist through the IRC, and the participant was prescribed mood stabilizers and sleep medicine due to her self-reported insomnia.

Due to her high-risk status, in addition to safety checks during regularly scheduled CETA sessions, the counselor followed up first with twice-daily home visits and gradually via phone calls. During this time, the counselor initiated CETA treatment with the participant. They worked on initial components of behavioral activation and collaboratively generated a list of activities, such as praying, cleaning the house and walking around the house, that could help her regulate her mood, as well as use strategies for relaxation to manage acute distress.

After completing six CETA sessions, she showed significant improvement in her trauma and depression symptoms and no longer reported suicidal thoughts. She actively participated in general activities, such as going to the farm with her family, gardening, walking around the neighborhood, singing, going to the mosque, visiting her cousin's house, playing guitar, helping others and volunteering. At the end of 12 CETA treatment sessions, she no longer reported sleeping problems, was consistent with her medications and attended regular appointments with her doctors to successfully control her hypertension and diabetes. She was able to manage her household responsibilities, sold snacks for income and reported improved connection and closeness with her family.

## Case 3: Karen – 45-year-old female with hypertension

At the initial safety assessment, this participant presented with 'low' suicide risk, stating, '*I want to die*' but no other safety concerns. However, her risk was elevated to 'moderate' due to additional reporting during her first CETA session, including thoughts and plans to kill herself by taking poison. She also presented with post-traumatic stress and depression symptoms, reported sleep problems (staying up too late thinking about her life problems and not having skills to manage it) and low motivation to adhere to her routine hypertension medication. She was living with her nine children and six grandchildren in a small house, with the youngest being seven years old, and she made ends meet by selling vegetables in the community and tending to farm animals. She expressed being 'very distressed' and 'ready to die' due to the pressure of earning for her family, while still grieving the death of her husband. Due to her chronic trauma history, she had been previously receiving psychosocial support and psychotropic medication through IRC services.

The client reported recent urges to kill herself, as well as several physical symptoms, including physical fatigue and headaches. In her CETA sessions, she frequently cried about her problems and reported feeling lonely and hopeless about her life. She developed an increased risk for suicidality with homicidal ideation toward her 14-year-old daughter, whom she suspected of having an affair with an older man. The client frequently stated unhelpful thoughts such as, '*I am a useless mother*' and '*my daughters are having bad lives because I am useless*'. As part of her safety plan, the counselor created a 3-day safety contract and several new activities for behavioral activation, including visiting friends and neighbors, going for leisure walks with her children, listening to religious talks and praying at the mosque.

This participant received 16 CETA treatment sessions, acquiring skills to manage her feelings of helplessness, solve problems (e.g., finding work to support herself and her family financially) and seek support and resources (such as from her prayer groups at the mosque). By the end of treatment, she showed greater hopefulness about her future, along with improvements in her mental health and medication adherence. She reported no longer having suicidal or homicidal thoughts or plans and was regularly taking her hypertension and psychotropic medications. She had successfully worked with social services to regain full custody of her daughter, who was no longer in the situation with the older man. Additionally, she reported using several CETA skills, including behavioral activation to strengthen her social supports, relaxation techniques to manage sleep problems, problem-solving skills to develop financial contingencies and effective coping skills.

## Building provider confidence and capacity requires ongoing supervision and support

While CETA's structured apprenticeship approach (Murray et al., 2011) provided a foundation in clinical care, many newly trained counselors initially expressed discomfort managing high-risk cases. This discomfort included uncertainty about how to prioritize safety steps, anxiety about potential negative outcomes and concern over managing emotional responses from participants while also liaising with other non-health agencies (such as case managers) for their other unmet needs that may be adding to their mental health distress. Through regular supervision and skill-building, the counselors gained confidence in conducting risk assessments, developing safety plans and engaging family or community support. Supervision sessions provided a structured space to discuss challenging cases, reflect on decision-making and practice applying the safety protocol in simulated scenarios. Counselors also received real-time guidance from the local CETA supervisors during follow-up visits, which helped them navigate difficult situations and reinforced adherence to the protocol. The safety follow-up sessions were particularly crucial for high-risk cases, as they helped demonstrate to participants that the counselors are caring for and helping them, and *that they are not alone*. These experiences suggest that, in addition to risk management, navigating these implementation challenges highlighted the importance of ongoing support and adaptive problem-solving, which may inform future implementation efforts in similar contexts. Furthermore, we learned that this protocol may have therapeutic relevance and was perceived to be meaningful by the participants.

## Discussion and lessons learned

This study provides insights from the implementation of a structured suicide safety protocol for displaced adults living with chronic physical health conditions in a humanitarian setting. Despite global recognition of suicide prevention as a public health priority (UN SDG, 2023; WHO, 2025), practical strategies for operationalizing suicide risk management in complex, low-resource environments remain limited (IASC, 2022; UNHCR, 2023). Our experience offers key lessons that may be relevant for researchers, program implementers and policymakers working in similar contexts.

### Context-adapted protocols facilitate clinical decision-making and suicide risk assessment

While suicide risk was relatively common in the settlement (we identified nearly 6% of the study participants at elevated risk) among participants (nearly 6%), no standardized screening or response protocol existed within the settlement's existing psychosocial services to address this risk. By adapting the CETA safety protocol and aligning it with existing referral pathways – including IRC-supported primary care clinics, psychosocial crisis teams and, where appropriate, community and religious leaders – we developed a structured suicide risk management process that could be sustained in this setting. While in line with documented limitations of clinical judgment, self-reports and formal risk scales (Large et al., 2022), our approach emphasized structured protocols, supervision and multiple data sources – including collateral information (e.g., from CETA counselors, supervisors and in consultation with the trainer) and behavioral indicators (e.g., treatment response, such as symptom remission, improvement in attendance to CETA sessions and following safety steps of the safety protocol) – to guide risk management decisions. Tailoring delivery to local resources, including leveraging community health workers when family or social support was absent, may support the feasibility of implementing such a protocol and, in this study, appeared to show acceptability at the participant level. These observations also reinforce recommendations that suicide prevention could be embedded within routine, multidisciplinary care to maximize accessibility and reduce stigma (Ahmedani et al., 2014; Leaune et al., 2020). We acknowledge that the 'safety risk' approach in suicide prevention is contested, with prior reviews questioning its predictive validity (Large et al., 2022). In our setting, the protocol functioned as a structured framework to facilitate decision-making, supervision and referral in the absence of existing standardized procedures rather than a predictive tool. We view our protocol's

utility as pragmatic – supporting consistency and accountability in care – rather than a predictive measure of future behavior.

Overall, we learned how suicide safety protocols can be implemented as part of existing health services, even in resource-constrained, high-vulnerability settings such as the setting in which our study was conducted. Our findings contribute to the limited but growing evidence base for operationalizing suicide prevention within humanitarian health programming and highlight practical strategies for adapting protocols to local realities. Future efforts should prioritize building sustainable referral pathways, addressing stigma and investing in provider training to strengthen suicide prevention efforts within global health and humanitarian response frameworks.

## Recommendations

Implementing suicide safety protocols is a feasible and pragmatic first step in managing concomitant safety risks in patients with chronic diseases in humanitarian settings.

Comprehensive data collection and monitoring are key steps in this safety protocol as they are essential for assessing ongoing mental health services and guiding quality improvement initiatives (Betz et al., 2019). In the context of this study, routine screening for suicide risk helped identify participants in need of targeted support, and monitoring behavioral and clinical indicators informed individualized follow-up. Based on our experience, tracking challenges encountered by counselors – such as discomfort, workflow interruptions or logistical barriers – can help programs iteratively refine protocol delivery and strengthen staff support mechanisms. Routine screening for suicide risk among patients with chronic illnesses may enable early identification and intervention.

Moreover, monitoring healthcare utilization patterns and outcomes could facilitate adaptive planning and resource allocation (WHO, 2025).

## Conclusions

While there are limited data available on the prevalence of suicidality and mental health problems among displaced Myanmar communities, it is well-documented that displaced populations are at a higher risk of mental health problems. Nongovernmental organizations and international agencies providing mental health and psychosocial supports to displaced populations might consider incorporating safety protocols that are adaptable to local resources, respond to individual needs and provide program participants with tools to navigate their difficult circumstances and seek help when needed. Reflecting on the experiences of counselors and participants in implementing the protocol provided critical insight into real-world barriers and facilitators, which guided our adaptation, training and supervision strategies and may be relevant to other programmatic contexts. Further, recognizing and addressing the intersecting vulnerabilities of adults with chronic physical health conditions, many of whom are older adults, can forge pathways toward greater dignity, resilience and well-being for this often-marginalized segment of society. Given the increased global focus on addressing suicidality, it is advisable to implement a safety protocol that identifies and keeps people safe until more services can be identified for referral.

**Open peer review.** To view the open peer review materials for this article, please visit http://doi.org/10.1017/gmh.2025.10101.

**Supplementary material.** The supplementary material for this article can be found at http://doi.org/10.1017/gmh.2025.10101.

**Data availability statement.** The data that support the findings of this study are available from the corresponding author upon reasonable request.

**Author contribution.** All authors who participated in the development of this manuscript contributed substantially. The corresponding author (SMS) has the permission of the co-authors to act on their behalf to confirm the accuracy and integrity of the work, and to proceed with this submission.

**Financial support.** This study was supported by the Enhancing Learning and Research for Humanitarian Assistance (ELRHA) under the Research for Health in Humanitarian Crises (R2HC) funding call. SMS was supported by the Global Mental Health (GMH) Training Program (Grant Number: T32MH103210) funded by the NIMH of the National Institutes of Health.

**Competing interests.** The authors declare none.

**Ethics statement.** Ethical approval for the trial was obtained from Khon Kaen University (IRB 00008614).

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
