## [Reviewer Report]

This case study paper seeks to describe the implementation of the Common Elements Treatment Approach (CETA) suicide safety protocol as part of an intervention trial in northern Thailand for displaced adults from Myanmar with chronic physical health conditions. The paper addresses the neglected topic of suicide among displaced persons and the link with chronic conditions is novel. However, I don’t feel the case study contains sufficient analytical depth to warrant publication. The journal guidance notes that ‘Case studies should contain critical evaluation of the projects, well-developed discussions that provide constructive information, and recommendations for improvement within the field’. I’m afraid I don’t see how this case study paper does this. Instead, it just describes the safety protocol that was used, the number requiring follow-up assessment, and then presents three cases of people with the interaction of suicide risk with diabetes and hypertension. It does not critically evaluate this work or present a detailed analysis of it.

In addition, while the authors reflect on lessons learnt, it is not clear how these lessons reflect what is presented in the findings. For example, where are the findings on cultural sensitivity, feasibility, and acceptability? Similarly, the process is not adequately described on how the suicide protocol was adapted with community members to ensure on cultural sensitivity, feasibility, and acceptability.

These weaknesses are a real shame as the subject matter is important and neglected.

---

## [Reviewer Report]

Dear authors,

thank you very much for this interesting manuscript on this important, critical topic. In general, the presentation of the suicide safety protocol is presented in a clear, comprehensible and understandable way.

However, a more detailed presentation of the overall structure of the project is missing, as well as a more in-depth discussion of any difficulties and challenges that may exist in this approach.

In the following, I have some detailed feedback/questions

Key Message:

- „within routine primary care“ – as I understood, the presentation of the cases did not take place in routine primary care but within the context oft he CETA intervention? This should be corrected. However, the transferability of this safety protocol to primary care could well be discussed,

- „evidenced based suicide prevention“ –please integrate relevant previous work into the introduction

- „In any context, there may be particularly vulnerable populations (such as ethnic minorities) who may require additional considerations to ensure to adoption of suicide risk reduction strategies.“ – this has not been discussed, or more, it has been presented how it is used in the context of a camp for displaced adults, which are vulnerable population. Please review this key message

Abstract

- „We describe the screening, risk assessment and treatment action planning for high-risk suicide cases“ – what differentiates your description from the trials that have used the protocol before?

- „including challenges faced“ – this has not really been the topic of discussion

Introduction

- In humanitarian emergencies, a particularly vulnerable group are individuals living with chronic illnesses (Schmid and Raju, 2021). – please provide more literature/data as this is the focus of this manuscript

- „…(CETA) approach that has been used in numerous trials and a range of humanitarian settings (see for example Bogdanov et. al, 2021; Murray, et al., 2020; Bonilla-Escobar, et.al 2018). CETA safety was integrated as part of a mental health program for displaced adults from Myanmar with chronic physical health conditions.“ – any information about the previous experiences with this protocol? It seems it has been used but no literature about it, which seems unlikely to me.

- „Read more about the context in the region here: https://www.hius.

- org/en/news/mae-la-refugee-camp-in-thailand-a-difficult-place-to-be-a-child“ – please write what is relevant fort he manuscript, don’t provide a link

- „We adapted the CETA safety protocol delivery to be in line with existing resources and referral systems.“ – please provide more details and rationale about the adaptations

- „procedures for the other risk categories“ – please mention

Results

- please provide a brief but more detailed description of the overall context, i.e. the study in which the integration of the saftey plan took place. Please mention which participants were included and how the assessment oft he physical illnesses took place

- „There was no relapse to suicidality“ – within which time frame? Were there planned follow-ups?

Lessons learned

- „interdisciplinary collaboration“ – please share more details what structures/support systems were present, and which also maybe not

- „cultural sensitivity“ – important topic. However, this needs a deeper discussion, as it seems that the choice of different counsellors were the only intercultural adaptation. Did the participants chose the counsellors? Or who decided, which counsellor treated which participant?

- How did community involvement take place?

- „The safety follow-up sessions were particularly crucial for high-risk“ – there was nothing written about follow up appointments in the descriptions, while I agree that this would be an important aspect. Please add details.

Best regards

---

## [Reviewer Report]

The paper, Integrating a Safety Protocol into Programming for Vulnerable Populations with Chronic Physical Health Conditions: A Case Study among Displaced Myanmar Adults in Thailand, provides valuable insights into the application of the CETA approach and safety protocol for moderate- to high-risk clients within vulnerable populations. It is engaging, well-written, and concisely presented. However, it would benefit from a more detailed discussion of the implementation process, community reception, on-the-ground challenges, and broader implications for displaced adults with comorbid conditions.

The background section is clear, concise, and well-written, effectively addressing key issues and providing sufficient context for the study. Adding some more detail on the CETA approach and safety protocol in the final paragraph would improve clarity.

The program context section effectively outlines the broader setting and details of the parent trial. The program adaptation and implementation section provides an overview of the protocol, though further elaboration is needed. Additionally, the specific location/setting where the program was implemented is unclear and should be explicitly stated.

Table 1: The phrase “want to commit suicide” should be reconsidered for precise and sensitive language.

Table 2: Offers a useful overview of safety risk levels.

In the results section, further clarification is needed regarding the screening process: Who was screened? How were participants selected? Were there any individuals who refused to participate? This is not clear.

Regarding CETA adaptation, translation is mentioned, but translation alone does not constitute adaptation. More details are needed on how the program was tailored to the specific needs of the population.

Participant voices are not prominently present—incorporating more quotes would provide valuable insight into their experiences. The narrative appears overly simplistic, lacking discussion of challenges encountered during implementation, the role of context, and consideration of factors such as client acceptability.

In the lessons learned section, the concept of “integration” needs further clarification—what did integration entail, and what challenges emerged? While the lessons presented are valuable, they focus on what worked rather than insights gained from challenges encountered.

The feasibility and acceptability of the program require a more rigorous assessment—on what basis is this conclusion drawn? The absence of participant experiences and on-the-ground implementation details makes it difficult to evaluate these claims.

Additionally, the paper should engage with the documented challenges in suicide risk assessment, including the limitations of clinical judgment, the unreliability of self-reports, and the poor performance of risk scales (Large et al., 2022). How did these issues influence the implementation of the risk assessment protocol and safety plan?

Finally, more reflection on the broader implications of the findings is needed to strengthen the paper’s contribution.

---

## [Editor Report]

Dear Prof Sardana,

Thank you for submitting this incredibly interesting case study! It is important to understand further the implementation of such programmes in settings underrepresented in the literature. As you can see, the reviews varied greatly. However, I would be open to receiving a revision of this manuscript, if you feel you can sufficiently address the reviews. 

One point in addition to those made by the reviewers is that while the case studies themselves are important, I would like to see a greater emphasis on suicide and suicide prevention, given the scope of the special issue. This may include greater description of the effects of the programme on suicidal thoughts and behaviours, and more.

Thank you and all the best,

Dr. Sandersan Onie

---

## [Reviewer Report]

Thank you for the revisions. This manuscript is a significant improvement, and the authors have taken into consideration the reviewers’ comments. The context of the study and the study procedures are now described in greater detail, and additional information on the implementation processes has been provided.

However, the manuscript would benefit from a more substantive reflection on key challenges encountered during the implementation of the protocol. For example, there is brief mention of counsellors’ discomfort in implementing the protocol, but this point is only lightly addressed. The manuscript does not explore how such barriers were navigated. These kinds of insights are critical for readers, as they can inform future implementation efforts in similar contexts.

If the word count permits, the authors could incorporate these reflections within the results section, potentially as part of the case study description, to provide a more nuanced understanding of the on-the-ground realities of implementation.

---

## [Editor Report]

Dear Dr Sardana,

Thank you very much for your revisions to the manuscript. They are substantial, and I believe this is important work to share. 

I have provided further comments, given that only one peer reviewer could provide comments in this round. 

1. Since you have updated your WHO GHE reference to 2025, please update the estimates accordingly. E.g., the latest Suicide Worldwide found that 73% of suicides occured in LMICs and not 77% as was reported in the 2021 iteration. 

2. On page 8, it now states that the ethics was for diabetes and/or hypertension. Did the ethics also include mental health/suicidality. If so, please state to avoid confusion. 

3. Given that suicide screening has its issues - it needs to be even more clearly stated that anyone who sored more than 1 on the screening questions, thereby disclosing any thought or behaviour of suicide was provided with a safety plan. 

4. Similar to the above, the idea of safety risk is problematic in suicide prevention - as noted by a previous review. I would be transparent in this process, but minimize description and remove the table, as the results do not rely heavily on it. I encourage discussion surrounding its utility, given the current literature, and whether it should or should not be used in this context.

5. While the CETA process has been provided with more detail - given it is the foundation of this case study - I would suggest providing maybe 3-4 more sentences on what it specifically entails. 

6. I would urge the authors to temper the language in the discussion further, as the study, while providing some interesting case studies, may not fully provide enough information or data for the protocol to be considered ‘feasible’ or demonstrate ‘importance’. 

7. Given that this is a case study, the learnings should focus more on how the cases provided informed the approach, rather than broad learnings on suicide prevention in humanitarian settings as a whole, which would require more data to address. Indeed, it is fine to include discussion on suicide prevention in humanitarian settings; however, this should be limited as the data provided is only three cases. The greatest value of this manuscript may not lie not suicide prevention in broad humanitarian settings, but rather in how people responded, how it was received, and how this study can help other organizations implement similar protocols to reach and help each individual. Recommendations for broad suicide prevention in humanitarian settings based on three case studies may not be appropriate. 

8. Furthermore, on page 16, it states that we ‘learned how suicide safety protocols can be feasibly implemented...’ This is challenging to glean as the methods did not have sufficient detail for the reader to understand why it wouldn’t be challenging to implement in the first place. 

Once again - this is incredibly important work - I believe some edits should strengthen its impact. 

Thank you and all the best,

Dr. Sandersan Onie

---

## [Reviewer Report]

Congratulations to the authors for the substantial improvements made in the revised manuscript and for persisting through the review process. The reflections on implementation challenges are now much clearer, demonstrating the complexities involved. Importantly, reflections on strategies to mitigate these challenges adds value, offering concrete lessons for future studies to draw from. These enhancements have strengthened the manuscript, making it a more meaningful contribution to the field.

---

## [Editor Report]

Dear Prof Sardana,

Thank for revising the manuscript to address the comments and concerns. It now reads much better and is clearer in its intention. Upon reading the comprehensively revised manuscript. However, I think that the revision may need to be implemented more thoroughly prior to publication, particularly surrounding the evidence and strength of argument that can be gleaned from case studies:

Currently the manuscript still has phrases such as:

“Context-adapted protocols are essential for identifying and managing suicide risk in humanitarian settings” - whereas the information presented in the manuscript does not substantiate this claim

“In our trial, suicide risk was..” - the current paper does not report a trial, and discussion of the larger trial in a paper which focuses on the case studies may not be appropriate.

Please go through the manuscript carefully and ensure claims and recommendations pertain to the study at hand. 

Furthermore, this was in the discussion:

"many newly trained counselors initially expressed discomfort managing high-risk cases. This

discomfort included uncertainty about how to prioritize safety steps, anxiety about potential negative

outcomes, concern over managing emotional responses from participants while also liaising with

other non-health agencies (such as case managers) for their other unmet needs that may be adding to

their mental health distress." - this is new information regarding the study, that was not introduced in the results section. Lessons should be featured in the results, with the discussion focusing on the interpretation and implications of these findings. 

Finally, the manuscript could benefit from a deeper typographical reading. For example, in text referencing is inconsistent in some places e.g., WHO vs World Health Organization. 

As previously mentioned, this is an important contribution to the literature, and I hope you will consider revising it to ensure it has a higher probability for impact. 

Thank you and all the best,

Dr. Sandersan Onie

---

## [Editor Report]

Dear Prof Sardana,

I thank the authors for their revisions, and feel this is an important contribution to the literature. I am happy now to recommend acceptance and with the authors the best of luck in their further work. 

Thank you and all the best,

Dr. Sandersan Onie